# Stage-differentiated ensemble modeling of DNA methylation landscapes uncovers salient biomarkers and prognostic signatures in colorectal cancer progression

Sangeetha Muthamilselvan☯, Abirami Raghavendran☯, Ashok Palaniappan ID *

Department of Bioinformatics, School of Chemical and BioTechnology, SASTRA Deemed University, Thanjavur, India

☯ These authors contributed equally to this work.
* apalania@scbt.sastra.edu

**Data Availability Statement:** https://doi.org/10.6084/m9.figshare.13013852.

## Abstract

### Background

Aberrant DNA methylation acts epigenetically to skew the gene transcription rate up or down, contributing to cancer etiology. A gap in our understanding concerns the epigenomics of stagewise cancer progression. In this study, we have developed a comprehensive computational framework for the stage-differentiated modelling of DNA methylation landscapes in colorectal cancer (CRC).

### Methods

The methylation β-matrix was derived from the public-domain TCGA data, converted into M-value matrix, annotated with AJCC stages, and analysed for stage-salient genes using an ensemble of approaches involving stage-differentiated modelling of methylation patterns and/or expression patterns. Differentially methylated genes (DMGs) were identified using a contrast against controls (adjusted p-value <0.001 and |log fold-change of M-value| >2), and then filtered using a series of all possible pairwise stage contrasts (p-value <0.05) to obtain stage-salient DMGs. These were then subjected to a consensus analysis, followed by matching with clinical data and performing Kaplan–Meier survival analysis to evaluate the impact of methylation patterns of consensus stage-salient biomarkers on disease prognosis.

### Results

We found significant genome-wide changes in methylation patterns in cancer cases relative to controls agnostic of stage. The stage-differentiated models yielded the following consensus salient genes: one stage-I gene (*FBN1*), one stage-II gene (*FOXG1*), one stage-III gene (*HCN1*) and four stage-IV genes (*NELL1*, *ZNF135*, *FAM123A*, *LAMA1*). All the biomarkers were significantly hypermethylated in the promoter regions, indicating down-regulation of expression and implying a putative CpG island Methylator Phenotype (CIMP) manifestation.

**Funding:** This work has been funded by DST-SERB grant EMR/2017/000470/BBM to A.P. (Department of Science & Technology - Science & Engineering Research Board, Govt. of India). The funders had no role in study design, data collection and analysis, decision to publish, or preparation of the manuscript.

**Competing interests:** The authors have declared that no competing interests exist.

A prognostic signature consisting of FBN1 and FOXG1 survived all the analytical filters, and represents a novel early-stage epigenetic biomarker / target.

## Conclusions

We have designed and executed a workflow for stage-differentiated epigenomic analysis of colorectal cancer progression, and identified several stage-salient diagnostic biomarkers, and an early-stage prognostic biomarker panel. The study has led to the discovery of an alternative CIMP-like signature in colorectal cancer, reinforcing the role of CIMP drivers in tumor pathophysiology.

## Introduction

Colorectal adenocarcinoma (CRC) is a major malignant disease with devastating incidence and mortality, being the cancer with the third highest global burden of disease, after lung and breast cancers, and accounting for 1.36 million new cases annually [1]. The etiology of CRC involves chromosomal instability (involving accumulation of mutations in oncogenes and tumor suppressor genes), microsatellite instability (MSI) (leading to loss of DNA mismatch repair) and CpG island methylator phenotype (CIMP), observed in nearly 85%, 15% and 10–40% respectively of all reported sporadic cases [2–4]. Epigenetic dysregulation is a key driver of these processes, and DNA methylation is the most important epigenetic modification [5, 6]. DNA hypomethylation could cause gain-of-function of oncogenes [7], and might aid severe tumor progression [8]. It has been found that large hypomethylation blocks are a universal characteristic of colorectal cancers and other solid tumors [9]. Hypomethylation could also contribute to tumor initiation and progression by a general increase in genomic instability [10]. DNA hypermethylation could cause loss-of-function of tumor suppressor genes, and hypermethylation in the germline could cause heritable loss of gene expression through genomic imprinting [11]. Aberrant hypermethylation of specific CpG islands has been observed to occur in colorectal cancer. The CpG island methylator phenotype (CIMP) was originally discovered in a subset of colorectal cancers [12], and subsequently refined to the involvement of five genes *CACNA1G*, *IGF2*, *NEUROG1*, *RUNX3*, and *SOCS1* [13]. Methylation changes contributing to phenotypic aberrations need not be localized to promoter regions but could occur in the gene coding regions and intron-exon structures [14–17]. The persistence of such modifications throughout the tumor cell lifetime has also been demonstrated by Lengauer et al. [18], who showed that methylation aberrations and genome instability were correlated, suggesting a key role for such aberrations in tumorigenic chromosomal segregation processes.

The Cancer Genome Atlas (TCGA) is a comprehensive resource of genome-wide mutation, expression and DNA methylation profiles of 46 different types of cancers [19]. Besides the TCGA, the International Human Epigenetic Consortium is devoted to data-driven understanding of the role of epigenomics in normal vs disease states [20]. Methylation patterns constitute an emerging class of promising prognostic factors mainly due to: (i) the persistence of widespread DNA methylation changes; (ii) the occurrence of such changes much ahead of the consequent changes in gene expression; and (iii) the ability to detect these changes in body fluids and blood plasma [21]. Few methylation markers have been previously translated to clinically applicable biomarkers [22], but it is known that tumorbehavior corresponds with epigenomic changes as reflected in differential DNA methylation [23]. Early detection may reduce the mortality rate via tailored adjustments to the treatment regimen, with the result of

fewer side-effects and better patient compliance. Chen et al., demonstrated a method to screen multiple types of cancer using a methylation-based blood test four years before conventional diagnosis [24]. A consensus approach to identifying significant methylation signatures in each stage of colorectal cancer progression would increase the utility and reliability of putative bio-markers. This motivated our interest in a systematic investigation of stage-salient epigenetic factors using several model-driven approaches, with the main objective of obtaining diagnostic and prognostic biomarkers.

## Methods

### Data preprocessing

Methylation data from 27k assays was used, since it is preferentially enriched in epigenetic pro-files in the proximal promoter regions (relative to 450k assays which are enriched in probes in the gene body and intergenic regions) [25]. Processed Level-3 27k CRC methylation data was retrieved from TCGA [26]. All samples in the dataset were processed and submitted by a single organization (namely 05: JHU_USC center), ensuring uniformity in data processing. MBatch analysis yielded low (<0.3) Dispersion Separability Criterion (measured as the ratio of between-batch dispersion to within-batch dispersion), indicating negligible batch effects and obviating the need for batch-correction (https://bioinformatics.mdanderson.org/public-software/mbatch/). The data containing the methylation β-values for each probe in each sam-ple was converted into a matrix with probes as rows and cases as columns. Each probe corre-sponds to one CpG site in the genome. A single gene may be under the control of multiple epigenetic sites, hence multiple probes may be associated with the same gene. It is noted that multiple probes usually exist for the same gene. The probes which have 'na' values were dis-carded from the analysis. To transform the range of methylation values from (0,1) to (-∞, +∞), we used the following function on the β-matrix values, to obtain the M-value matrix [27]:

$$M_i = \log_2[\beta_i/(1-\beta_i)] \tag{1}$$

In our study, two M-value matrices were considered: one, where all the probes were used in the analysis; and two, where the probes corresponding to one gene were represented by an average of their values ('avereps'), thus reducing the M-value matrix from a probe:sample matrix to a gene:sample matrix. Further, we filtered out the probes/genes showing little change in methylation (defined as $\sigma < 1$) across all cases in the M-value matrices. The latest clinical data (clinical.cases_selected.tar.gz) was obtained from the GDC by matching on the patient barcode [28]. The stages were annotated for both the β-matrix and M-value matrices using the 'Pathologic_stage' attribute encoded in the clinical data. Cases with unknown stage ('NA' val-ues) were discarded. The stage information was mapped to the American Joint Committee on Cancer (AJCC) Tumor-Node-Metastasis (TNM) classification system [29] (Table 1).

The final β and M-value matrices were subjected to stage-differentiated contrast analysis with a battery of six different methods, described below. All analysis was carried out on R [30].

### Modelling

To compensate for the assumptions specific to individual modelling approaches, an ensemble of models was explored.

**(1) Linear modelling with M-values.** Linear modelling is essential to identify linear trends in expression across cancer stages and thereby detect stage-sensitive patterns. We used

**Table 1.** AJCC cancer staging.

| TCGA Stage | TNM Classification | Cases | |
|---|---|---|---|
| I | T1N0M0 | 50 | |
| II | - | 17 | 86 |
| IIa | T3N0M0 | 64 | |
| IIb | T4aN0M0 | 5 | |
| III | - | 16 | 60 |
| IIIa | T1-T2N1/NcM0 | 3 | |
| | T1N2aM0 | | |
| IIIb | T3-T4aN1/NcM0 | 21 | |
| | T2-T3N2aM0 | | |
| | T1-T2N2bM0 | | |
| IIIc | T4aN2aM0 | 20 | |
| | T3-T4bN2bM0 | | |
| | T4bN1-N2M0 | | |
| IV | - | 35 | 36 |
| IVa | Any-T Any-N M1a | 1 | |
| CONTROL | - | | 42 |
| NA | - | | 1 |

The correspondence between the AJCC staging and the TCGA staging for COADREAD is noted. 'NA' denotes cases where the stage information is unavailable. Sample sizes are successively aggregated to the parent stage.

the R package limma [31] for linear modelling of stagewise expression using the complete M-value matrix, with multiple probes per gene (S1 Table in S1 Text).

**(2) Linear modelling with avereps matrix.** This is essentially similar to the above model, except that the input was the 'avereps' matrix, where the methylation of each gene was represented by the average of its M-values across all its probes (S2 Table in S1 Text). Such alternative representations of the methylation data negotiate a tradeoff with respect to information loss and interpretability.

In both the linear models, the controls contributed to the intercept of the design matrix, while the stages were represented as indicator variables [32]. The linear fit was subjected to empirical Bayes adjustment to obtain moderated t-statistics. These results were then used for the stage-differentiated contrast analysis

**(3) Association between methylation status and phentoype.** The strength of the association between the methylation levels of CpG sites and the phenotype of interest (CRC-stage) could enable the identification of relevant markers. We used the R package CpGassoc [33] to estimate this association based on ANOVA with multiple hypothesis correction. The β-matrix was used as input, and five factors (control, stage I, stage II, stage III, stage IV) were specified as the target phenotype.

**(4) The Chip Analysis Methylation Pipeline (ChAMP).** The Chip Analysis Methylation Pipeline (ChAMP) integrative analysis suite uses limma to identify differentially methylated probes (DMPs) from the β-matrix [34]. A mapping of sample IDs with the pathological stage phenotype was provided as an additional input file. In addition, the identification of differentially methylated regions (DMRs), consisting of polygenic genomic blocks, was performed using DMRcate in ChAMP (with preset p-value cutoff<0.05) [35]. GSEA was used to identify the enrichment of DMPs and DMRs in the MSigDB pathways [36], using the Fisher Exact test calculation with adjusted p-value < 0.05.

**(5) Correlation between gene methylation and expression.**   We used MethylMix2.0 to estimate the correlation between the methylation and actual expression patterns of each gene [37]. The expression data for the cases of interest were retrieved from TCGA (gdac. broadinstitute.org_COADREAD.Merge_rnaseqv2_illuminaga_rnaseqv2_unc_edu_Level_3_ RSEM_genes_data.Level_3.2016012800.0.0.tar.gz). MethylMix was executed with the preset correlation cutoff ($> |0.3|$), and statistical significance was assessed using Wilcoxon Rank Sum test with adj. p-value $< 0.05$.

**(6) Modelling expression from methylation.**   We used the R package BioMethyl to model the aggregate expression level of a gene from its methylation patterns [38]. The gene expression matrix was estimated using the methylation β-matrix and then subjected to linear modelling with limma, followed by stage-differentiated contrast analysis.

## Stage-differentiated contrast analysis

A directed two-tier set of contrasts was performed in limma to drill down to the stage-salient genes:

1. Tier I: Stage-differentiated contrast against controls. Four pairwise contrasts were performed, one for each of the stages I, II, III and IV. To identify reliable DMGs, the following criteria were used: |lfc M-value| $>2$, and adj. p-value $<0.001$.

2. Tier II: Inter-stage contrasts. Six pairwise contrasts between the stages (namely: I-II, I-III, I-IV, II-III, II-IV, and III-IV) were performed (p-value for each contrast: $<0.05$).

   To illustrate, a putative DMG identified in Tier I would undergo three inter-stage contrasts in Tier II, to ensure stage-salience. For example, a putative stage-II DMG established by Tier I, would have to pass the following inter-stage contrasts: stage-II vs stage-I, stage-II vs stage-III and stage-II vs stage-IV, for confirmation as stage II-salient DMG.

## Identification of stage-salient biomarkers

Finding the consensus of a set of methods with different algorithms overcomes the biases specific to individual methods, and enables screening out false positives. Consensus was obtained by finding the agreement among the results of the various methods used. At least three methods should agree on a given DMG's stage-salience, for confirmation as *consensus* stage-salient biomarker.

## Survival analysis

The survival data for each case was obtained from the following attributes encoded in the clinical data: patient.vital_status, patient.days_to_followup, and patient.days_to _death. The association between consensus stage-salient DMGs and case overall survival (OS) was evaluated by univariate Cox proportional hazards regression model using the R survival package [39]. This uncovered potential prognostic stage-salient genes from the methylation analysis, using a significance cutoff $< 0.05$. Such prognostic genes were used as the independent variables in a regression model to estimate the survival risk of each case. Based on this risk score, cases with colorectal cancer were categorized into high and low groups using the optimal cut point determined by the maxstat (maximally selected rank statistic) [40]. Kaplan-Meier estimation was then applied to the median survival times of these two groups for flagging significant differences, providing a prognostic assessment of the biomarkers of interest.

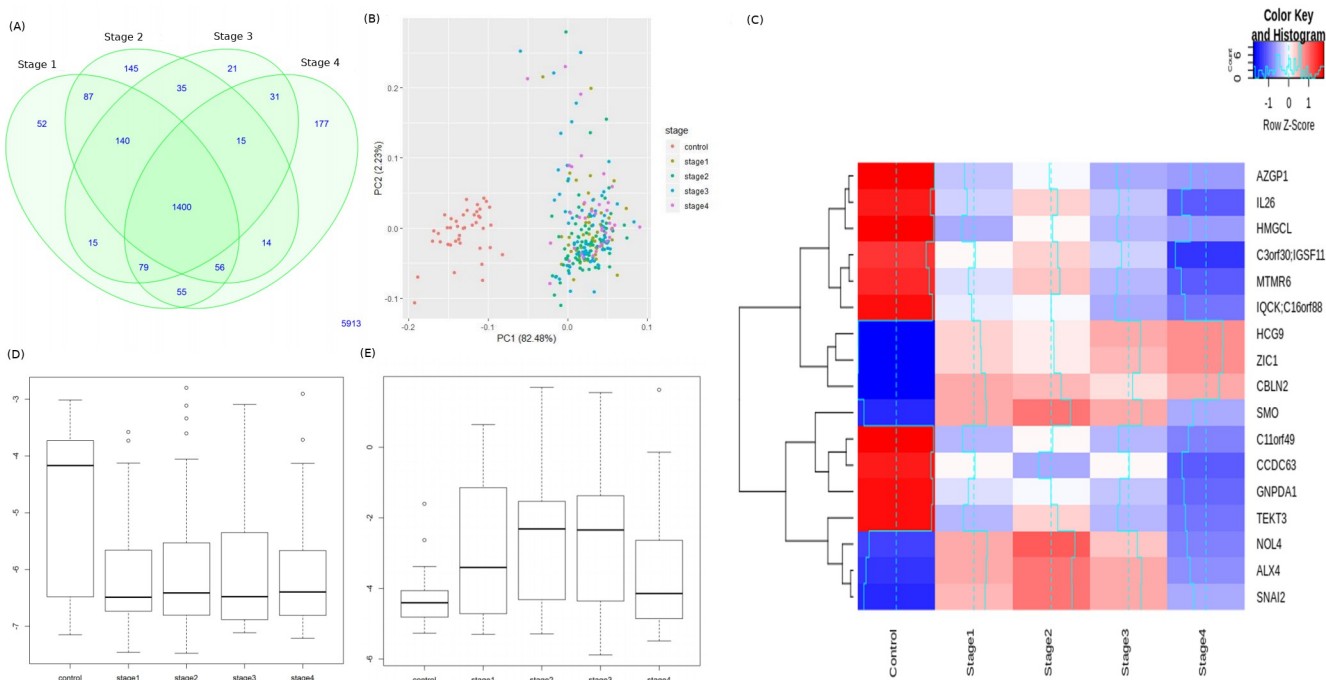

**Fig 1. Linear modelling with M-value matrix, all probes.** (A) Venn distribution of significant DM genes in each stage relative to control. (B) Distribution of samples based on the top two principal components of the top 100 genes shows a clear separation of cancer cases (labelled by stage) from controls. (C) Stagewise methylation portraits of the top four significant stage-specific DMGs. The contrast with the control is especially evident. Also shown are the stagewise methylation levels of (D) TMEM179, and (E) MEOX2.

## Results

### Linear modelling with M-values (at the probe-level)

The number of significant genes present in each stage-control pair from the Tier-I contrasts is shown in Fig 1A. Using the top 100 DM genes of the linear model (given in S3 File in S1 Text), we found a clear separation between controls and stage samples (Fig 1B). The top genes in each stage (by adjusted p-value of contrast with control) are shown in Table 2, with |lfc M-

**Table 2. Top ten genes of the linear model at the probe level.**

| ID | Stage I lfc ($\beta_1$) | Stage I lfc ($\beta_2$) | Stage III lfc ($\beta_3$) | Stage IV lfc ($\beta_4$) | Adj. P-val | Methylation status |
|---|---|---|---|---|---|---|
| GPR75-ASB3 | 2.28 | 2.19 | 2.16 | 2.32 | 1E-82 | Hyper |
| TM4SF19 | -3.63 | -3.58 | -3.72 | -3.71 | 1E-82 | Hypo |
| CNRIP1 | 2.74 | 2.60 | 2.68 | 2.97 | 1E-78 | Hyper |
| PDE4A | 1.68 | 1.58 | 1.60 | 1.71 | 1E-71 | Hyper |
| KRTAP11-1 | -2.36 | -2.30 | -2.37 | -2.40 | 1E-70 | Hypo |
| ADHFE1 | 3.15 | 2.97 | 3.00 | 3.43 | 1E-69 | Hyper |
| FAM123A | 3.56 | 3.18 | 3.43 | 3.90 | 1E-69 | Hyper |
| KHDRBS2 | 2.30 | 2.16 | 2.10 | 2.34 | 1E-68 | Hyper |
| AJAP1 | 2.52 | 2.44 | 2.46 | 2.64 | 1E-68 | Hyper |
| NALCN | 2.96 | 2.80 | 2.94 | 3.25 | 1E-68 | Hyper |

The log fold-change of M-value of the probe in each stage relative to the controls, followed by p-value adjusted for the false discovery rate, and the methylation status of the gene in the cancer stages with respect to the control.

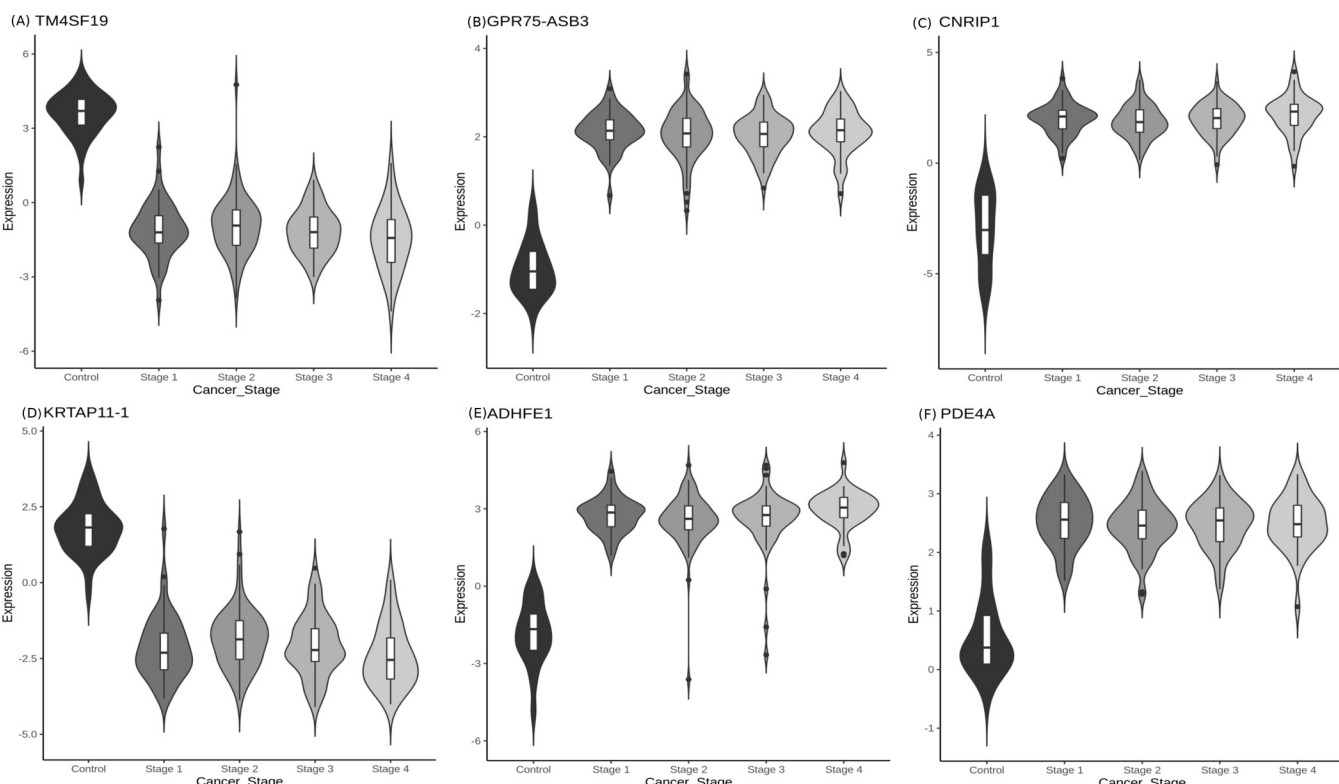

**Fig 2. Top DMGs identified from linear modelling.** (A) GPR75-ASB3, (B) TM4SF19, (C) CNRIP1, (D) KRTAP11-1, (E) ADHFE1 and (F) PDE4A. For each gene, notice that the trend in methylation could be either hyper-or hypo-methylation relative to the control. TM4SF19 and KRTAP11-1 are hypomethylated whereas CNRIP1, GPR75-ASB3, ADHFE1, PDE4A are hypermethylated.

value| and inferred regulation status. The top four genes of each stage were used to construct a stagewise methylation heatmap (Fig 1C). Fig 1D and 1E show boxplots of stagewise methylation levels for two representative genes: TMEM179, mutations in which could cause MSI [41]; and MEOX2 whose promoter methylation status is a known CRC marker [42]. The stagewise methylation patterns of the top linear model genes are shown in Fig 2. It is notable that a naturally occuring read-through fusion protein GPR75-ASB3 is the top linear model gene with significant differential expression in all stages relative to the control. GPR75-ASB3 is positively differentially expressed in the lung as well as different keratinocyte cell types, and evidence is emerging of its role in other cancers [43]. In this light, GPR75-ASB3 could play a significant role in colorectal cancers which are of epithelial origin. The top 100 significant stage-specific genes, listed in S3 File in S1 Text, were used in the consensus analysis.

## Linear modelling with avereps matrix (at the gene-level)

The methylation levels of genes with multiple probes were averaged using limma's avereps function, and summarized to one value. The number of genes present in each stage-control pair from the Tier-I contrasts is shown in Fig 3A. Using the top 100 genes of the linear model (given in S4 File in S1 Text), we found a clear separation between controls and stage samples (Fig 3B). The top genes in each stage (by adjusted p-value of contrast with control) are shown in Table 3, with |lfc M-value| and inferred regulation status. The top four genes of each stage were used to construct a stagewise methylation heatmap (Fig 3C). Fig 3D and 3E shows the boxplots of stagewise methylation levels for two representative genes, NALCN and GLRX.

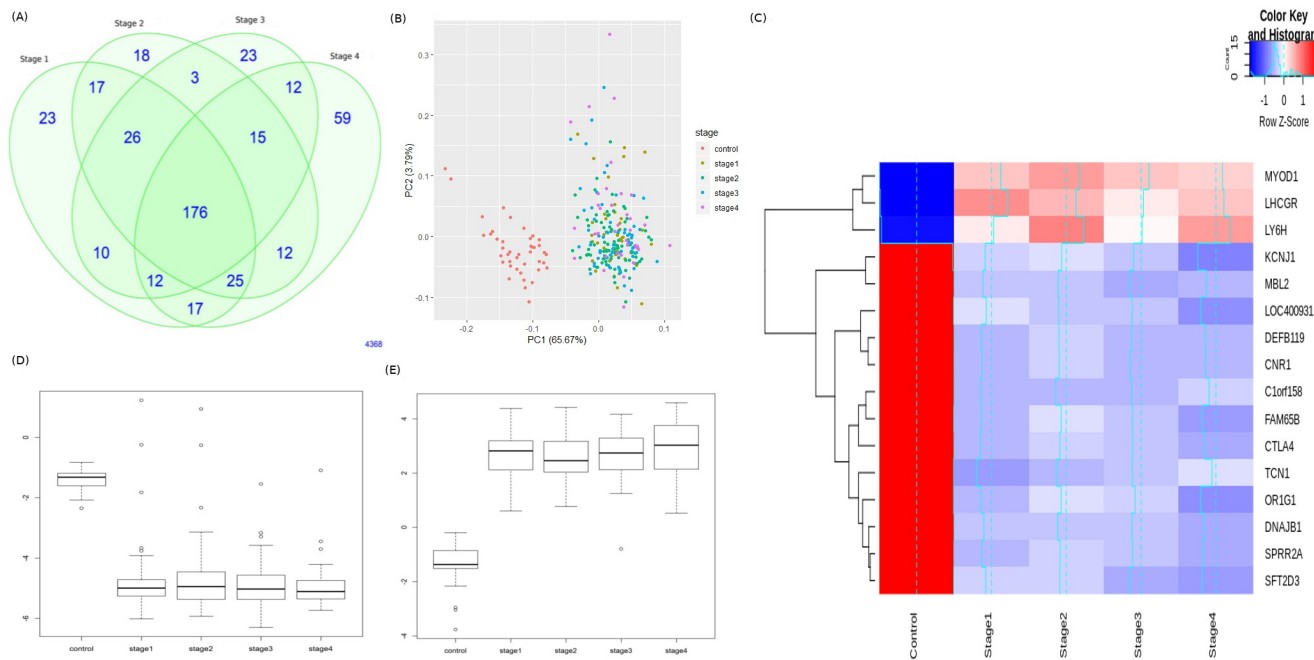

**Fig 3. Linear modelling with M-value matrix, avereps transformation.** (A) Venn distribution of significant DM genes in each stage relative to control. (B) Distribution of samples based on the top two principal components of the top 100 genes shows a clear separation of cancer cases (labelled by stage) and controls. (C) Stagewise methylation portraits of the top four significant stage-specific DMGs. The stark contrast with the control is especially evident. Also shown are the stagewise methylation levels of (D) NALCN, and (E) GLRX.

Mutations in NALCN have been reported in sporadic CRC [44]; here NALCN is seen to be significantly hypermethylated, indicating the same outcome (loss of function) could be effected in multiple ways. GLRX is a target of the activating transcription factor MEOX2 [45]. It is observed that *LY6H* showed both hypermethylation and hypomethylation when compared to the controls, indicating the role of experimentation necessary to clarify its role in colorectal cancer progression. The top significant 100 genes of each stage, listed in S4 File in S1 Text, were used for the consensus analysis.

**Table 3. Top ten genes of linear modelling with averaging of multiple probes.**

| ID | Stage I lfc ($\beta_1$) | Stage I lfc ($\beta_2$) | Stage III lfc ($\beta_3$) | Stage IV lfc ($\beta_4$) | Adj. P-val | Methylation status |
|---|---|---|---|---|---|---|
| TM4SF19 | -3.63 | -3.57 | -3.72 | -3.71 | 1E-82 | Hypo |
| GPR75-ASB3 | 2.28 | 2.19 | 2.15 | 2.32 | 1E-82 | Hyper |
| CNRIP1 | 2.74 | 2.60 | 2.67 | 2.97 | 1E-77 | Hyper |
| KRTAP11-1 | -2.36 | -2.30 | -2.38 | -2.40 | 1E-70 | Hypo |
| ADHFE1 | 3.15 | 2.96 | 2.99 | 3.43 | 1E-69 | Hyper |
| FAM123A | 3.56 | 3.18 | 3.42 | 3.89 | 1E-68 | Hyper |
| AJAP1 | 2.53 | 2.44 | 2.46 | 2.64 | 1E-67 | Hyper |
| NALCN | 2.96 | 2.79 | 2.95 | 3.25 | 1E-65 | Hyper |
| IRF4 | 1.99 | 1.83 | 1.89 | 2.13 | 1E-65 | Hyper |
| PRKAR1B | 3.38 | 3.13 | 3.24 | 3.50 | 1E-65 | Hyper |

The log fold-change of M-value of the gene in each stage (relative to the control) is given, followed by p-value adjusted for the false discovery rate and the methylation status of the gene in the cancer stages with respect to the control. A consistent methylation pattern is observed for all the top genes.

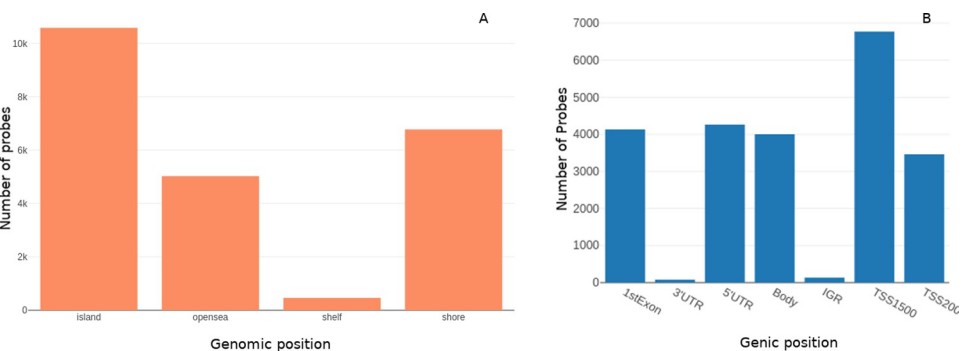

**Fig 4.** Distribution of probes based on (A) genomic position: opensea, shore, island, shelf; (B) gene context: transcription start site (TSS), exons, untranscribed regions (UTRs), and intergenic regions (IGR).

## Association with phenotype

The ANOVA from CpGassoc yielded p-values and log fold-changes, which were used to identify significant genes for each stage using the criteria given in Methods. The top 100 genes of each stage from this analysis (given in S5 File in S1 Text) were used for the consensus investigation.

## DMP analysis with ChAMP

The summary features of the β matrix dataset were evaluated using ChAMP (Fig 4). The DMPs were identified using ChAMP analysis from the β matrix. All the inter-stage contrasts yielded null results (i.e, no significant genes), except for stageII–stageIV contrast. Due to this, the top 100 DMPs from the stage vs control contrasts were used for the consensus analysis directly. Contrasts that showed significant DMPs were subjected to a further DMR analysis, to enable identification of DM genes. The stage-salient DMR regions (genes) determined are provided in S6 File in S1 Text, and summarized in Table 4. The stage-II vs stage-IV DMR contrast yielded three genes, namely PLAG1, SOCS2, and NNAT. It is observed that these genes might be critical players in the transition to malignancy. Interestingly, some genes were differentially methylated in all the stagewise contrasts with the control; such genes are differentially methylated agnostic of stage and could serve as valuable drug targets for CRC therapy. The top such genes included EYA4, WT1, DCC, RP11, GATA4, MSX1, DLX5, BNC1, WT1-AS, and ZIM2. A total of 31 such genes were identified and tabulated in S7 Table in S1 Text. The DMPs and DMRs from the analysis were subjected to GSEA and these results could also be found in S6 File in S1 Text. Fig 5 shows representative DMP and DMR plots using ChAMP.

**Table 4. Contrast-wise counts of DM probes and DM regions.**

| Contrast | DMPs | DMRs |
|---|---|---|
| Control and Stage 1 | 11045 | 34 |
| Control and Stage 2 | 11254 | 35 |
| Control and Stage 3 | 11254 | 36 |
| Control and Stage 4 | 11108 | 34 |
| Stage 2 and Stage 4 | 404 | 3 |

No DM regions were found for the contrasts not shown, namely the stage-pairs: [(1,2), (1,3), (1,4), (2,3), (3,4)].

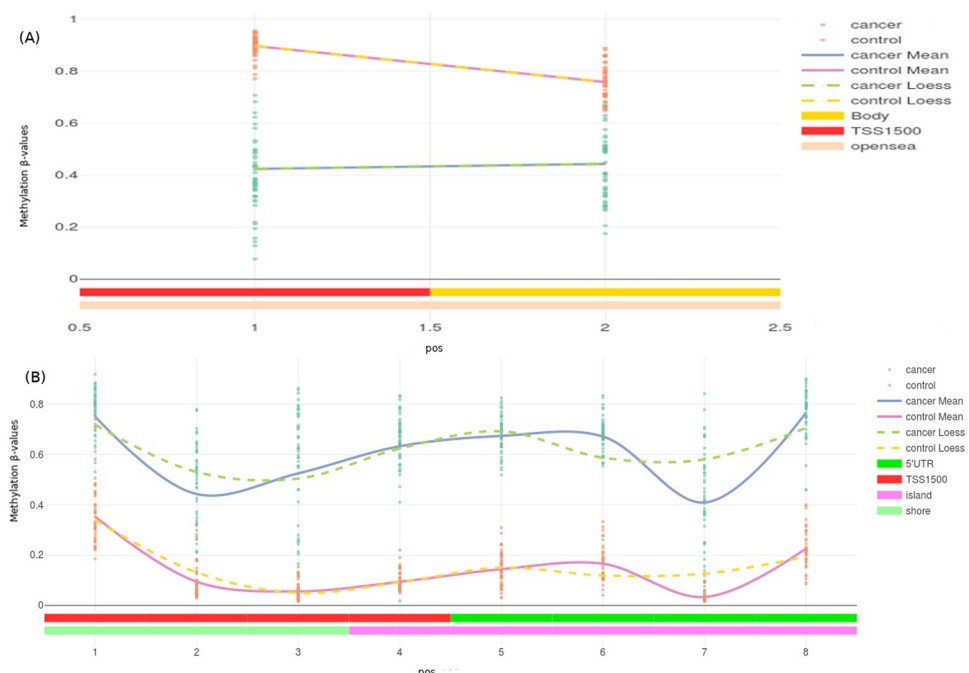

**Fig 5. DMP and DMR plots using ChAMP.** (A) DMP plot of FCN2 for stage-I vs control illustrating significant hypomethylation (B) DMR plot of transcriptional activator EYA4 for stage-I vs control illustrating significant hypermethylation. Solid lines represent mean values while dashed lines represent the loess.

## Methylation and expression correlation analysis

Differential methylation (DM) calculated from stage vs control contrasts ranged from -0.7 to +0.8, and genes could be hyper- or hypo-methylated based on the sign of the DM value. There were 209, 441, 275, and 134 driver genes in each of the contrasts with the controls (stage-I, stage-II, stage-III and stage-IV, respectively). All between-stages contrasts yielded null DM genes. The results from this analysis, including driver genes for all the contrasts, are provided in S8 File in S1 Text. It is notable that the top genes from an overall cancer vs control comparison included GATA4, CCDC88B, and WAS. Top 100 genes from each comparison with the controls were taken forward for consensus analysis. Certain genes emerged common to all the four comparisons with the controls, thereby suggesting stage-agnostic differential methylation events. The top such stage-agnostic differentially methylated genes included *CCDC88B*, *C1orf59*, *CHFR*, *ZP2*, *HOXA9*, *ELF5*, *FAM50B*, *MUC17*, *TBX20*, and *VSIG2*. Stage-agnostic genes hold promise as therapeutic targets for the treatment of colorectal cancer; the complete set of 56 stage-agnostic genes identified in this analysis is provided in S9 Table in S1 Text. Mixture models of genes, indicative of the number of methylation states, were constructed using MethylMix, and illustrated for a few stage-IV driver genes in Fig 6. The estimated correlation between the methylation levels and actual gene expression for the same genes shows the inverse relationship between methylation and gene expression, thereby highlighting the effect of epigenetic events (Fig 6).

## BioMethyl analysis

The significant stage-specific DEGs identified by BioMethyl are shown in UpSet plot [46] (Fig 7), and provided in S10 File in S1 Text. Top 100 genes of each stage from this analysis were taken forward for consensus analysis.

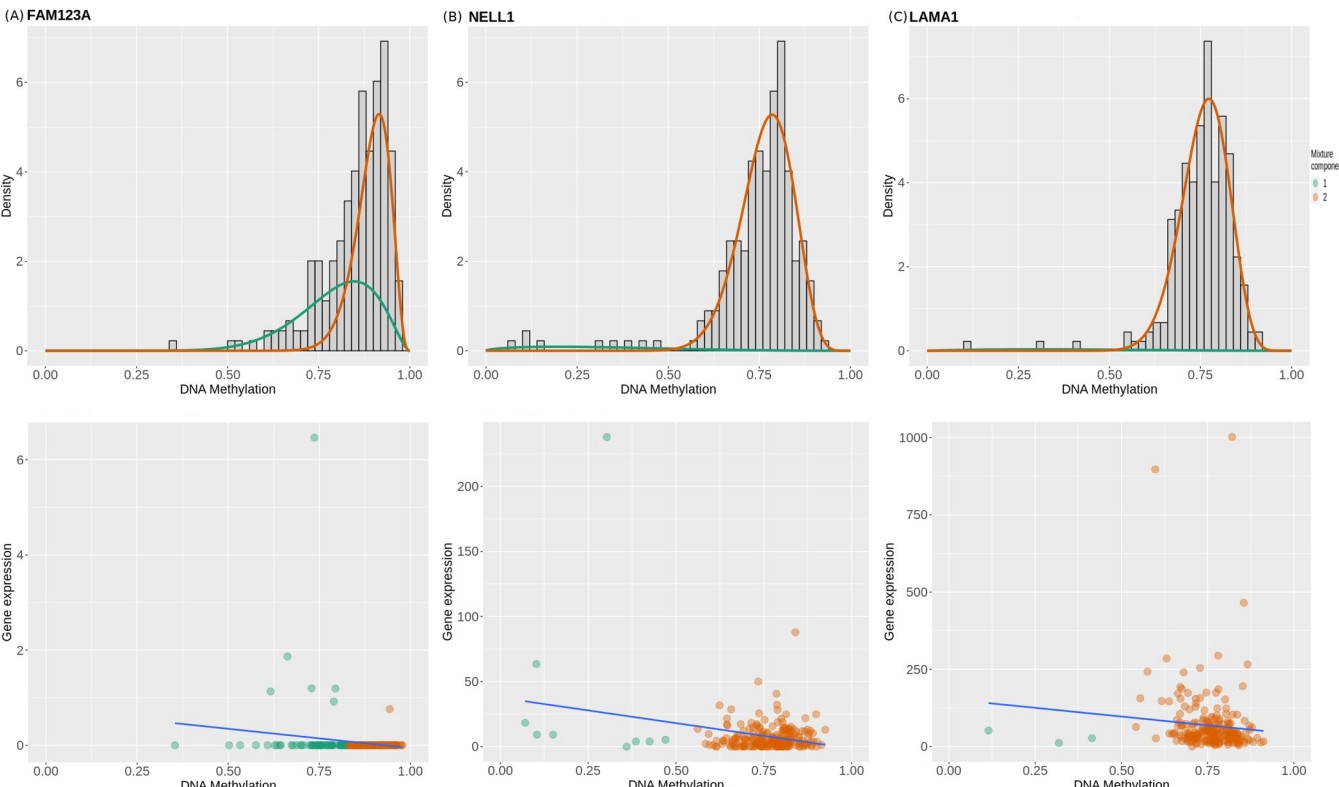

**Fig 6.** Mixture models and Correlation plots for (A) FAM123A, (B) LAMA1 and (C) NELL1. The x-axis indicates the level of methylation (in terms of β values); y-axis, the frequency. Mixture component curves represent density fits of the histogram. A negative correlation between methylation and expression is evident, indicating that methylation acts to repress gene transcription, though the strength of the inverse correlation varies from gene to gene. olour indicates the mixture model fit.

### Stage-salient consensus biomarkers

The top 100 significantly differentially-expressed genes of each stage from all the methods discussed above (collated in S11 File in S1 Text) were used for the consensus determination. The consensus analysis yielded seven stage-salient DMGs: one stage-I gene *(FBN1)*, one stage-II gene *(FOXG1)*, one stage-III gene *(HCN1)* and four stage-IV genes *(NELL1, ZNF135, FAM123A, LAMA1)*. Each of these stage-salient genes presented an |lfc M-value| > 0.4 with respect to the other stages, validating their salience. Fig 8 represents violin plots of the consensus biomarkers, and Table 5 presents a summary of the consensus analysis. Gene ontology (GO) analysis [47] of the consensus biomarkers yielded processes related to structural integrity of cell division processes, immunity dysfunction, and cell migration (Table 6). Detailed GO results are presented in the S12 File in S1 Text.

### Survival analysis

We constructed independent prognostic models of the stage-salient genes and identified the prognostically significant biomarkers as FBN1, FOXG1, HCN1, and LAMA1. The corresponding univariate Kaplan-Meier plots are shown in Fig 9. Rational combinations of stage-salient genes, termed ColoRectal cancer Signatures (CRS), were modelled using multivariate Kaplan-Meier regression, to yield a risk score. Risk scores were then used to estimate survival-effect significance, as described in Methods. The results of this exercise are summarised in Table 7. We found that CRS12 signature (consisting of FBN1 and FOXG1) yielded significant risk

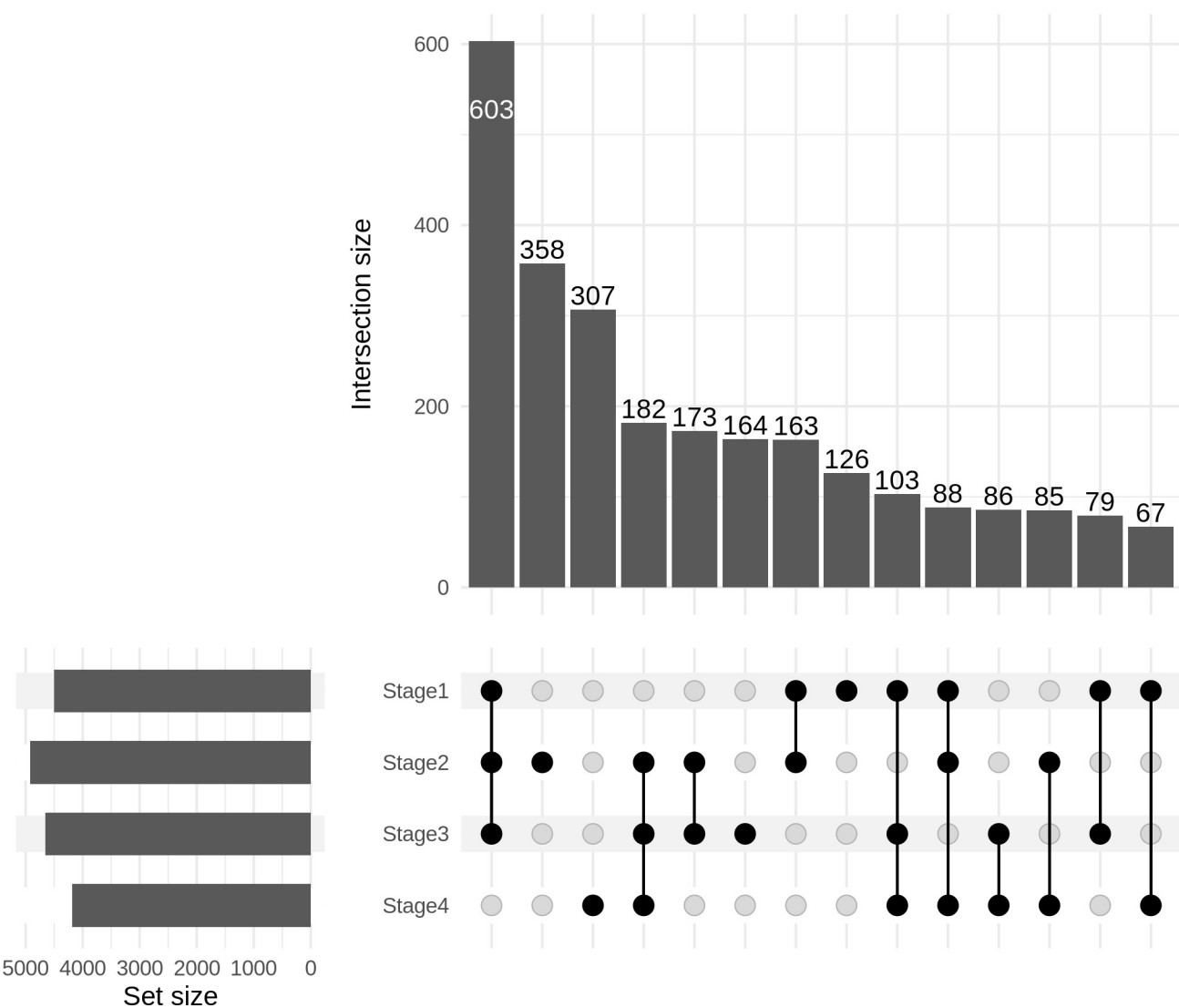

**Fig 7. UpSet plot of BioMethyl-based stagewise gene expression modelling.** The intersection of all stages yielded 3268 genes, which represent consistently differentially regulated genes.

scores in the multivariate Kaplan-Meier analysis, and both CRS12 and CRS34 (consisting of HCN1, NELL1, ZNF135, FAM123A, LAMA1) were significant in estimating overall survival (prognosis p-value ≤ 0.02) (Fig 10). S13 File in S1 Text provides survival plots of all possible signatures. At the end of our analysis pipeline, CRS12 passed all the filters and emerged as a significant early-stage panel for CRC prognosis.

## Discussion

CRC development is due to the accumulation of genetic and epigenetic changes of which DNA methylation is of paramount importance. DNA methylation profiles of colorectal cancer have been investigated in several previous studies using various approaches [48, 49]. It is well-known that changes in methylation status correspond with CRC progression [50]. Here we have designed a comprehensive approach to systematically analyze stage-differentiated DNA methylation patterns in colorectal cancer and their relationship to patient survival. Our study

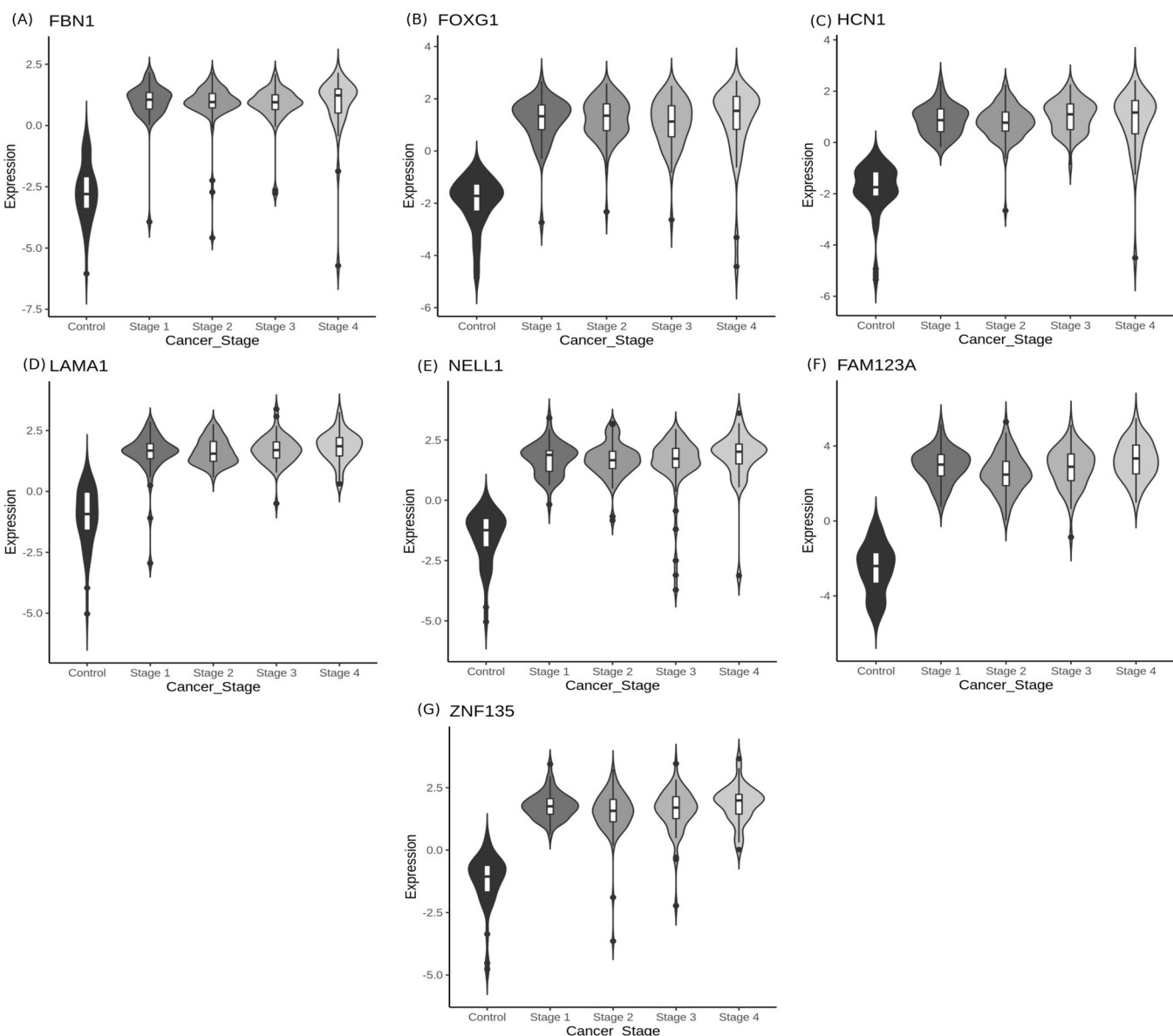

**Fig 8. Violin plots of stage-salient genes.** (A) Stage-I Gene FBN1, (B) Stage-II Gene–FOXG1, (C) Stage-III Gene–HCN1 and Stage-IV genes (D) LAMA1, (E) NELL1, (F) FAM123A, (G) ZNF135.

has yielded consensus stage-salient significantly differentially methylated genes, and evaluated their prognostic value. Corollary insights obtained in the course of our investigations, such as stage-agnostic genes, have been documented, and would also be of interest to researchers in the field. It is significant that none of the stage-salient genes figure as a cancer gene or a hallmark gene in the Cancer Gene Census [51]; HCN1 is notably marked as a candidate cancer gene based on mouse insertional mutagenesis experiments [52]. The dominant differentially methylated CpG site in all the stage-salient genes is located within the core / proximal promoter regions (Table 8). Mixture models of methylation levels of stage-salient genes, along with their inverse correlation to corresponding expression levels are shown in Fig 11, and unambiguously establish the epigenetic impact of the changes in methylation. Our findings are

**Table 5. Stage-salient biomarkers.**

| HGNC ID | Gene Name | Methods in agreement | Salience | Meth. status | Statistical significance | | | |
|---|---|---|---|---|---|---|---|---|
| | | | | | M value | Avereps | Cox analysis | Kaplan Meier |
| 3603 | FBN1 | Avereps, ChAMP | I | Hyper | 0.310 | 0.040 | 0.036 | 0.025 |
| 3811 | FOXG1 | Mvalue, Avereps, ChAMP, Methylmix | II | Hyper | 1E-16 | 0.003 | 0.019 | 0.037 |
| 4845 | HCN1 | Mvalue, Avereps, ChAMP | III | Hyper | 1E-17 | 0.022 | 0.031 | 0.059 |
| 7756 | NELL1 | Mvalue, Avereps, ChAMP | IV | Hyper | 1E-68 | 0.061 | 0.283 | 0.27 |
| 12919 | ZNF135 | Mvalue, ChAMP, Methylmix | IV | Hyper | 1E-76 | 0.062 | 0.096 | 0.084 |
| 26360 | FAM123A | Mvalue, ChAMP, Methylmix | IV | Hyper | 1E-115 | 0.097 | 0.30 | 0.28 |
| 6481 | LAMA1 | Mvalue, ChAMP, Methylmix | IV | Hyper | 1E-86 | 0.297 | 0.052 | 0.051 |

The results of the consensus analysis and univariate survival analysis are summarized. All the biomarkers showed hypermethylation, reflecting downregulation of gene expression.

further discussed in the context of the existing literature, and lead us to detect a strange CpG island methylator phenotype (CIMP) signature in colorectal cancer.

## Stage-salient DMGs

Promoter hypermethylation of FBN1, a glycoprotein component of calcium-binding extracellular matrix microfibrils [53], is a recognized biomarker of CRC [54, 55]. Our analysis supports this literature, while pinpointing the stage I-salience in its action. FOXG1 is well-known as an etiological factor in certain neurological disorders and plays a role in the epithelial-mesenchymal transition of CRC cells (a key hallmark of cancer progression), and is known to be overexpressed in CRC cases [56]. It is a nodal gene, with connections to oncogenic pathways like WNT pathway in hepatocellular carcinoma [57] and TGF-β pathway in ovarian cancer [58] Interestingly, FOXG1 was found to be a hypermethylated stage-II salient gene. HCN1, coding for hyperpolarization-activated cyclic nucleotide-gated channel subunits, is associated with low survival rates in breast, brain, and colorectal cancer [59]. We have identified HCN1 as a stage-III hypermethylated gene, suggesting a loss-of-function mechanism for its tumorigenic potential.

Our study has provided clear evidence that hypermethylation of LAMA1 (which codes for α-laminin of the extracellular matrix) is a stage IV-specific signature. Experimental evidence

**Table 6. GO analysis of stage-salient genes in the order of decreasing significance (i.e, increasing p–value).**

| GO ID | Term | Ontology | p-value |
|---|---|---|---|
| GO:1990047 | Spindle matrix | CC | 0.0001 |
| GO:0030109 | HLA-B specific inhibitory MHC class I receptor activity | MF | 0.0003 |
| GO:0032396 | Inhibitory MHC class I receptor activity | MF | 0.006 |
| GO:0042609 | CD4 receptor binding | MF | 0.0012 |
| GO:0032393 | MHC class I receptor activity | MF | 0.0013 |
| GO:0050930 | Induction of positive chemotaxis | BP | 0.0016 |
| GO:0050927 | Positive regulation of positive chemotaxis | BP | 0.0033 |
| GO:0050926 | Regulation of positive chemotaxis | BP | 0.0034 |
| GO:0008608 | Attachment of spindle microtubules to kinetochore | BP | 0.0043 |
| GO:0007094 | Mitotic spindle assembly checkpoint | BP | 0.0044 |

Ontology could be Cellular Compartment (CC), Molecular Function (MF), or Biological Process (BP).

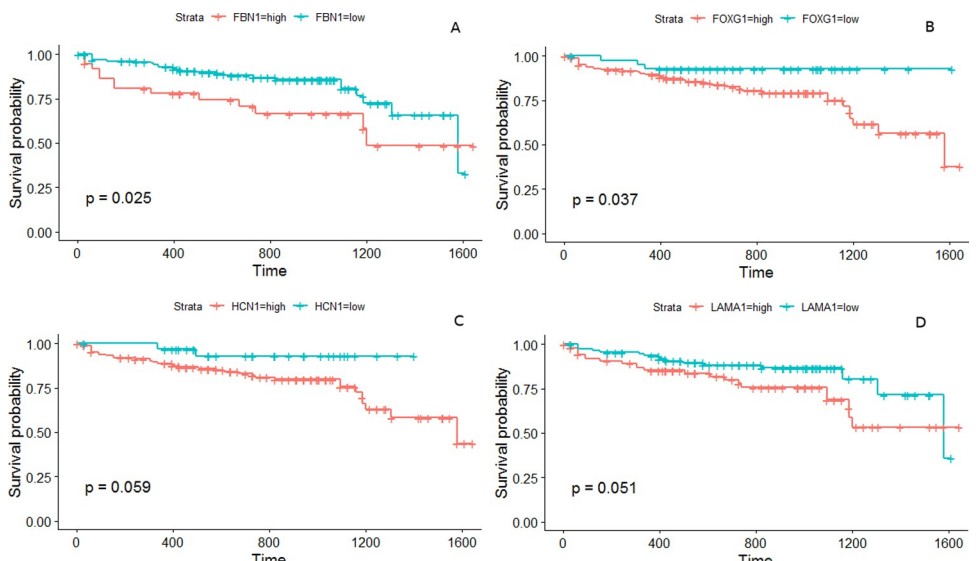

**Fig 9. K-M plots for the prognostically significant stage-salient genes.** (A) FBN1, (B) FOXG1, (C) HCN1, and (D) LAMA1.

for the hypermethylation of the promoter region of LAMA1 in CRC cases is available [60]. NELL1 is a known tumor suppressor gene [61], whose hypermethylation is associated with poor survival outcomes [62]. Here it is found to be a stage IV-specific hypermethylated gene, resonating with the above findings. ZNF135 is a zinc-finger protein involved in regulation of cell morphology and cytoskeletal organizations. Its expression and epigenetic regulation have been reported to be key in cancers of the cervix and esophagus, respectively [63, 64]. Here we have found that epigenetic silencing of ZNF135 is a key feature of stage-IV CRC. It is interesting that another member of the zinc-finger protein family, ZNF726, has been recently identified as the only methylated gene significantly associated with OS in patients with CRC,

**Table 7. Summary of selected multivariate prognostic models.**

| Signature | Stages | Biomarker | Weight | P-value | |
|---|---|---|---|---|---|
| | | | | Multivariate model | Prognosis |
| CRS12 | I+II | FBN1 | -0.62 | **0.015** | **0.005** |
| | | FOXG1 | -1.05 | | |
| CRS34 | III+IV | NELL1 | 0.1 | 0.172 | 0.02 |
| | | ZNF135 | -0.21 | | |
| | | FAM123A | -0.23 | | |
| | | LAMA1 | -0.39 | | |
| | | HCN1 | -1.1 | | |
| CRS234 | II+III+IV | FOXG1 | -0.99 | 0.0877 | 0.032 |
| | | HCN1 | -1.07 | | |
| | | NELL1 | -0.10 | | |
| | | ZNF135 | -0.22 | | |
| | | FAM123A | -0.37 | | |
| | | LAMA1 | -0.27 | | |

Weight denotes the coefficient in the multivariate model. The ultimate significant signature is highlighted.

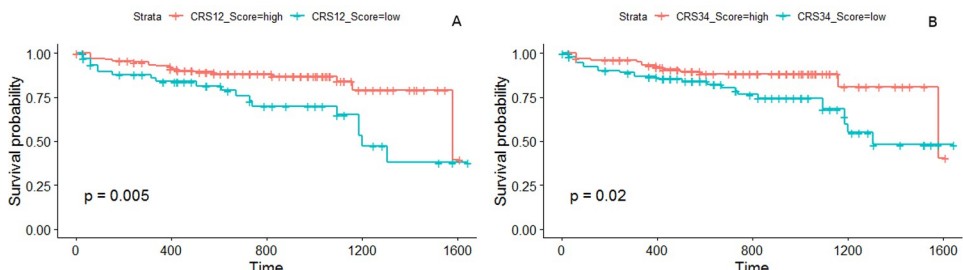

**Fig 10. Survival analysis of combination biomarker panels shows significance.** (A) Early-stage panel, CRS12; and (B) Late-stage panel, CRS34.

without regard for pathologic stage [65]. FAM123A, also known as AMER2, is associated with microtubule proteins [66], and is a paralog of the well-documented FAM123B, a tumor-suppressor whose loss-of-function by mutation, methylation and copy-number aberrations is known to play a pivotal role in colorectal cancer, especially in older patients [67–69]. It is significant that our study has uncovered FAM123A as a hypermethylated stage IV-specific DMG, signalling the need for experimental investigations. There is very little literature on the cancer significance of any of the above stage-salient genes, marking our findings as novel and important in the context of gaps in our knowledge.

## Putative CIMP signature

Aberrant methylation of CpG promoter regions causes stable repression of transcription leading to gene-silencing [70, 71]. In the context of tumorigenic processes, this is likely to lead to loss-of-function of tumor-suppressor genes. Multiple CpG islands might be methylated simultaneously in some cancers, paving the way for CpG island methylator phenotype (CIMP), first discovered in colorectal cancer [72]. CIMP is characterised by hypermethylation of CpG islands surrounding the promoter regions of genes involved in cancer onset and progression [73]. The phenotype is heterogenous with the type of tumor [74] and dependent on definition [75]. Table 8 suggests that the stage-salient hypermethylated biomarkers identified in our study are components constituting an aggregate novel CIMP, and there is preliminary experimental evidence in this direction. Earlier studies have identified LAMA1 as a CIMP panel constituent [50, 60]. FBN1 has been used as an epigenetic biomarker in diagnostic panels associated with CIMP-positive tumors [54, 76]. While this paper was under review, FAM123A has been used in a five marker panel to detect stage-IV CRC using blood samples [77]. The original CIMP had been associated with advanced T staging (T3/T4) [78], which accords with

**Table 8. Location of the major DM CpG site in stage-salient genes.**

| Stage-salient gene | DM CpG site | Distance to TSS | Location in the promoter region |
|---|---|---|---|
| FBN1 | cg18671950 | 146 | Proximal |
| FOXG1 | cg10300684 | 36 | Core |
| HCN1 | cg06498267 | 298 | Proximal |
| NELL1 | cg17371081 | 179 | Proximal |
| ZNF135 | cg16638540 | 144 | Proximal |
| FAM123A | cg22029275 | 73 | Core |
| LAMA1 | cg07846220 | 133 | Proximal |

All the hypermethylated CpG sites of stage-salient DMGs were found in the core/proximal promoter regions.

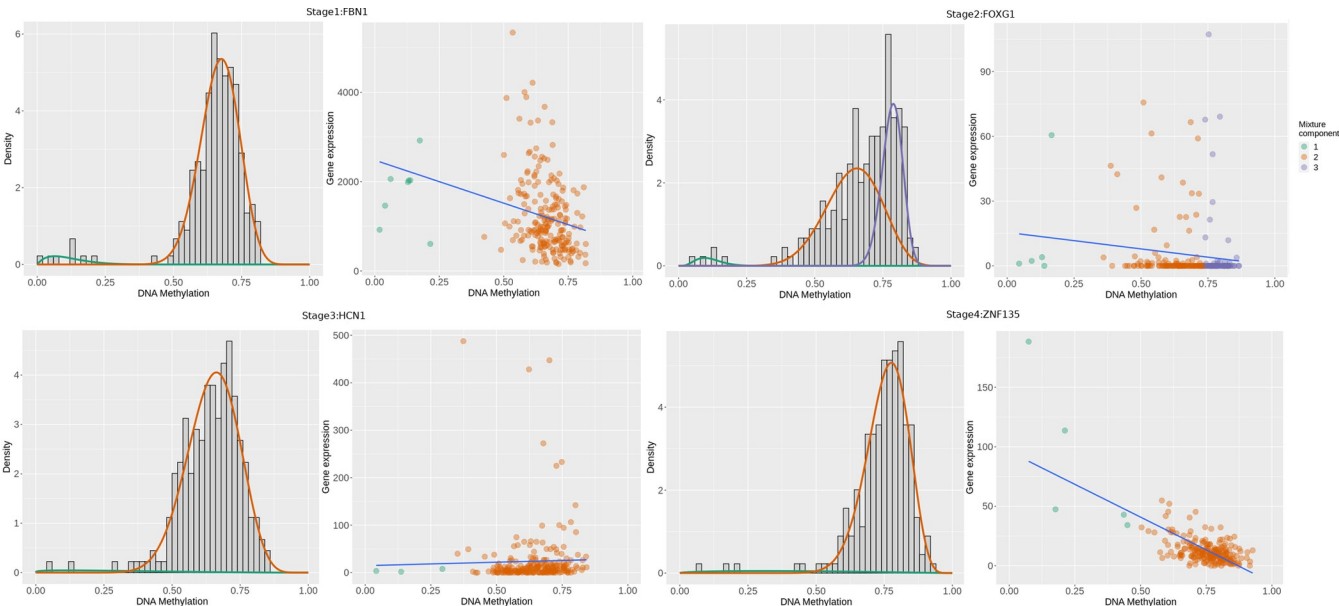

**Fig 11. Mixture models and correlation plots of stage-salient genes.** Shown are FBN1, FOXG1, HCN1, and ZNF135. Two mixture components are seen for FBN1, HCN1, and ZNF135, and three for FOXG1. A strong inverse correlation exists for all genes, except HCN1. Other stage-salient genes are shown in Fig 6.

our finding of four hypermethylated stage IV-salient DMGs. The biomarkers from our study contributing to the putative CIMP were tested with a standard survival analysis workflow yielding significant prognostication power for five of the seven stage-salient genes (Table 5). A Cox multivariate analysis of biomarker panels uncovered two signatures, an early-stage CRS12, and a late-stage CRS34 that might be prognostically valuable. In particular, CRS12 (composed of FBN1 and FOXG1) suggests a significant early-stage biomarker panel (p-value < 0.01) for the effective prognosis and stage-sensitive detection of colorectal cancer.

Diagnostic biomarkers that are also superior in prognostication power imply methylation events that are vital to tumor-specific pathophysiology. This suggests future directions for therapeutic intervention. Epigenetic intervention for CIMP-positive cancers has been advanced as a possible treatment strategy [79]. The alternative CIMP-like biomarkers could serve to stratify the cancer subtype, thereby facilitating precision medicine. The current standard of CRC screening is colonoscopy, an invasive method with a significant rate of complications. A non-invasive method based on molecular diagnostics would improve patient satisfaction and efficiency. Several studies have been conducted to identify and/or validate biomarkers for CRC diagnosis. It is recognized that DNA methylation patterns could serve as valid biomarker candidates [80, 81]. Freitas et al., have validated the performance of a 3-gene biomarker panel for the detection of colorectal cancer irrespective of the molecular subtype [82]. However optimal stage-salient epigenetic biomarkers have not yet been reported. Using hypermethylated DNA patterns as cancer markers offers the advantage of providing small targets with high concentrations of CpG for assays, useful for the design of analytical amplicons [83]. Hypermethylation in the gene body and upstream control regions like enhancers and insulators might affect transcription differently than hypermethylation of promoter regions [84, 85]. Further DNA methylation patterns in noncoding RNA genes seem to be important in tumorigenesis and progression [86]. Non-coding RNAs themselves play a significant role in epigenetic modification through the phenomenon of RNA-directed DNA methylation [48]. The nuanced relationship between methylation and gene transcription signals the need for clinical validation of our

results, however ensemble approaches such as the one used here suffer less uncertainties with respect to translation of the identified biomarkers. Since methylation mediates a direct epigenetic regulatory mechanism used by all life [87], it is hoped that the workflow herein designed would advance our understanding of the complex effects of methylation events, patterns, and landscapes in different settings, including in the developmental stages of life.

## Conclusion

We have developed a comprehensive computational framework for the consensus identification of stage-differentiated significant differentially methylated genes, and evaluation of their prognostic significance. Our analysis has yielded seven stage-salient genes, all hypermethylated in the promoter regions and relatively unreported in the literature: one stage-I gene *(FBN1)*, one stage-II gene *(FOXG1)*, one stage-III gene *(HCN1)* and four stage-IV genes *(NELL1, ZNF135, FAM123A, LAMA1)*. Stage-salient genes could serve as diagnostic biomarkers, and their concordant hypermethylation would signal a distinct CIMP-like character possibly promoting epigenetic destabilisation, which in turn would drive the progression of colorectal cancer. These findings lend further evidence to CIMP drivers of colorectal cancer and point more generally to a pervasive role for these aberrations in tumor biology that remains to be discovered. Independent prognostic evaluation of the stage-salient markers yielded significance for FBN1 and FOXG1. Survival analysis of biomarker signatures composed of the stage-salient genes yielded a significant early-stage panel consisting of FBN1 and FOXG1. Our studies have also spawned secondary results such as stage-agnostic genes that could serve as targets for drug discovery in CRC therapy. Consensus approaches, like the one used here, are more reliable, and the epigenetic biomarkers identified in our study could potentially advance the accurate early detection of colorectal cancers, their treatment and prognostic evaluation. The methods are extendable to the investigation of epigenomics in other cancers, normal/disease conditions, and developmental biology.

## Supporting information

**S1 Text.**
(TXT)

## Acknowledgments

We are grateful to the School of Chemical and Biotechnology, SASTRA Deemed University for computing and infrastructure support.

## Author Contributions

**Conceptualization:** Ashok Palaniappan.

**Data curation:** Abirami Raghavendran.

**Formal analysis:** Sangeetha Muthamilselvan, Abirami Raghavendran.

**Funding acquisition:** Ashok Palaniappan.

**Investigation:** Sangeetha Muthamilselvan, Ashok Palaniappan.

**Methodology:** Abirami Raghavendran, Ashok Palaniappan.

**Project administration:** Ashok Palaniappan.

**Resources:** Ashok Palaniappan.

**Software:** Sangeetha Muthamilselvan, Abirami Raghavendran, Ashok Palaniappan.

**Supervision:** Ashok Palaniappan.

**Validation:** Sangeetha Muthamilselvan, Ashok Palaniappan.

**Visualization:** Sangeetha Muthamilselvan, Abirami Raghavendran, Ashok Palaniappan.

**Writing – original draft:** Sangeetha Muthamilselvan, Ashok Palaniappan.

**Writing – review & editing:** Ashok Palaniappan.

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
