## [Decision Letter · Decision Letter 0]

7 Jul 2021

PONE-D-21-08604

Stage-differentiated ensemble modeling of DNA methylation landscapes uncovers salient biomarkers and prognostic signatures in colorectal cancer progression

PLOS ONE

Dear Dr. Palaniappan,

Thank you for submitting your manuscript to PLOS ONE. After careful consideration, we feel that it has merit but does not fully meet PLOS ONE’s publication criteria as it currently stands. Therefore, we invite you to submit a revised version of the manuscript that addresses the points raised during the review process.

We look forward to receiving your revised manuscript.

Kind regards,

Alessandro Weisz

Academic Editor

PLOS ONE

Additional Editor Comments:

The manuscript has been reviewed by two experts in the filed that both found it quite good and of interest, despite some problems, highlighted in particular by R.2, that must be addressed before it can be considered for publication.

Journal Requirements:

Reviewers' comments:

Reviewer's Responses to Questions

**Comments to the Author**

1. Is the manuscript technically sound, and do the data support the conclusions?

Reviewer #1: Yes

Reviewer #2: Partly

2. Has the statistical analysis been performed appropriately and rigorously? 

Reviewer #1: Yes

Reviewer #2: Yes

3. Have the authors made all data underlying the findings in their manuscript fully available?

Reviewer #1: Yes

Reviewer #2: Yes

4. Is the manuscript presented in an intelligible fashion and written in standard English?

Reviewer #1: Yes

Reviewer #2: No

5. Review Comments to the Author

Reviewer #1: In this paper Muthamilselvan et al., developed a comprehensive computational framework for stage-differentiated modelling of DNA methylation landscapes in CRC, found significant changes and discovery a novel CIMP-like signature

bearing potential clinical significance.

The data are supported by a strong statistical analysis.

The paper can be acceptted for pubblication.

Minor point

+Page 3 the word "Tomczak" must be deleted.

Reviewer #2: In this study, the Authors propose a computational workflow for the analysis of the DNA methylation aberrations in colorectal cancer (CRC) with a stage-differentiated perspective. Data have been collected from The Cancer Genome Atlas (TCGA) portal; as a result, the Authors identify 7 stage-characteristic genes that could be indicative of a novel CpG island methylator phenotype (CIMP).

The manuscript provides an interesting perspective and a detailed explanation of how DNA methylation data could be analyzed to detect epigenetic signatures between normal and tumoral samples or in different stages of the disease. Bioinformatic procedures are well described and depicting a useful workflow when handling big and complex data from repositories such as TCGA.

Unfortunately, however, I cannot avoid pointing out that the work has serious shortcomings that do not allow to accept its publication in the present version and must be mandatorily corrected.

Importantly, even if interesting, data are presented in a confused manner; in particular, the Authors should better define the aim of the study and the experimental design, carefully organize the Results, improve the Discussion and avoid exaggerated conclusions, concerning in particular inferences drawn from data that should be better supported by experimental validation. Specific comments are listed below.

Major Comments:

1. The Authors should carefully revise the text and correct some grammar mistakes.

2. Figures are in many cases blurred and not legible; their quality must be improved.

3. The Authors should pay attention to some typing mistakes and font differences (see, for example, pages 14 and 12).

4. TCGA collects data from at least 10 different studies on CRC; at least the cohort from which data have been collected should be reported in Materials and Methods.

5. TCGA collects data from 236 patients profiled with Human Methylation Bead Chip HM27, and 393 with HM450, measuring 27,000 and 480,000 CpG sites, respectively; why only the HM27 data have been used?

6. The correlation analyses between methylation and gene expression data are providing interesting information but should be better described by focusing the attention not only on the methodological procedure used but also on the biological meaning of the observed results.

7. In the Conclusions the Authors write: “All the stage-salient genes were found to be hypermethylated, indicating a novel CIMP-like character possibly promoting epigenetic destabilisation, which in turn would drive the progression of colorectal cancer”. First, it is not clear where the hypermethylation associated with these stage-salient genes is located (promoter, TSS, CpG island or gene body); this should be better explained. Then, the role of the stage-salient genes identified by the Authors should be better characterized in the context of CRC to indicate a possible novel CIMP-like phenotype; I would suggest the Authors to enrich the Discussion by adding more details and experimental evidence of the involvement of these genes in CRC pathogenesis.

6. PLOS authors have the option to publish the peer review history of their article (what does this mean?). If published, this will include your full peer review and any attached files.

Reviewer #1: No

Reviewer #2: No

---

## [Author Response · Author response to Decision Letter 0]

23 Jul 2021

>>>At the outset, we would like to thank the reviewers and the Editor for their valuable comments. 

The manuscript has been reviewed by two experts in the filed that both found it quite good and of interest, despite some problems, highlighted in particular by R.2, that must be addressed before it can be considered for publication.

this has been done. 

>>>We have now substantially revised the manuscript and expanded the scope of our investigations / discussion.

>>> done

>>>This has been done. 

Reviewer #1: In this paper Muthamilselvan et al., developed a comprehensive computational framework for stage-differentiated modelling of DNA methylation landscapes in CRC, found significant changes and discovery a novel CIMP-like signature

bearing potential clinical significance.

The data are supported by a strong statistical analysis.

The paper can be acceptted for pubblication.

Minor point

+Page 3 the word "Tomczak" must be deleted.

>>> Thank you. We would like to thank Reviewer #1 for their time and the kind comments. 

Reviewer #2: In this study, the Authors propose a computational workflow for the analysis of the DNA methylation aberrations in colorectal cancer (CRC) with a stage-differentiated perspective. Data have been collected from The Cancer Genome Atlas (TCGA) portal; as a result, the Authors identify 7 stage-characteristic genes that could be indicative of a novel CpG island methylator phenotype (CIMP).

The manuscript provides an interesting perspective and a detailed explanation of how DNA methylation data could be analyzed to detect epigenetic signatures between normal and tumoral samples or in different stages of the disease. Bioinformatic procedures are well described and depicting a useful workflow when handling big and complex data from repositories such as TCGA.

Unfortunately, however, I cannot avoid pointing out that the work has serious shortcomings that do not allow to accept its publication in the present version and must be mandatorily corrected.

Importantly, even if interesting, data are presented in a confused manner; in particular, the Authors should better define the aim of the study and the experimental design, carefully organize the Results, improve the Discussion and avoid exaggerated conclusions, concerning in particular inferences drawn from data that should be better supported by experimental validation. Specific comments are listed below.

>>> We would like to thank Reviewer #2 for the careful reading of our paper and the critical comments. We have addressed all the many valid points in the present revision. We have undertaken a major revision of the manuscript in line with the suggestions. 

1. The Authors should carefully revise the text and correct some grammar mistakes.

>>> yes

2. Figures are in many cases blurred and not legible; their quality must be improved.

>>> all figures have been redone in high-resolution tiff format

3. The Authors should pay attention to some typing mistakes and font differences (see, for example, pages 14 and 12).

>>> yes

4. TCGA collects data from at least 10 different studies on CRC; at least the cohort from which data have been collected should be reported in Materials and Methods.

>>> this has been identified and recorded in the manuscript under Methods. 

5. TCGA collects data from 236 patients profiled with Human Methylation Bead Chip HM27, and 393 with HM450, measuring 27,000 and 480,000 CpG sites, respectively; why only the HM27 data have been used?

>>> This point has also been addressed in the Methods section. Essentially 450k Chips show enrichment in gene body and intergenic regions. A distribution of the CpG sites with respect to the genomic / genic location clearly indicates this (data not shown). 27k data are enriched in CpG sites in promoter regions. Please refer the section under Methods. 

6. The correlation analyses between methylation and gene expression data are providing interesting information but should be better described by focusing the attention not only on the methodological procedure used but also on the biological meaning of the observed results.

>>> This has now been rectified. Indeed, we now show plots for only the stage-salient genes, to make the biological connections and meaning clear. We note that all the results from our investigations are available in the Supplementary Files. 

7. In the Conclusions the Authors write: “All the stage-salient genes were found to be hypermethylated, indicating a novel CIMP-like character possibly promoting epigenetic destabilisation, which in turn would drive the progression of colorectal cancer”. First, it is not clear where the hypermethylation associated with these stage-salient genes is located (promoter, TSS, CpG island or gene body); this should be better explained. Then, the role of the stage-salient genes identified by the Authors should be better characterized in the context of CRC to indicate a possible novel CIMP-like phenotype; I would suggest the Authors to enrich the Discussion by adding more details and experimental evidence of the involvement of these genes in CRC pathogenesis.

>>> We have now increased the literature weight for these statements and discussion. We have included a new Table 8 with the location of the Dm probes. and a new Figure 19 to support all these assertion. We also found a new publication citing stage-IV specificity for FAM123A while this manuscript was under review (medrxiv preprint of our work was available in October 2020). 

Thank you.

---

## [Editor Report · Decision Letter 1]

27 Aug 2021

PONE-D-21-08604R1

Stage-differentiated ensemble modeling of DNA methylation landscapes uncovers salient biomarkers and prognostic signatures in colorectal cancer progression

PLOS ONE

Dear Dr. Palaniappan,

Thank you for submitting your manuscript to PLOS ONE. After careful consideration, we feel that it has merit but does not fully meet PLOS ONE’s publication criteria as it currently stands. Therefore, we invite you to submit a revised version of the manuscript that addresses the points raised during the review process.

ACADEMIC EDITOR: Please insert comments here and delete this placeholder text when finished. Be sure to:

Indicate which changes you require for acceptance versus which changes you recommendAddress any conflicts between the reviews so that it's clear which advice the authors should followProvide specific feedback from your evaluation of the manuscript

We look forward to receiving your revised manuscript.

Kind regards,

Alessandro Weisz

Academic Editor

PLOS ONE

Journal Requirements:

Additional Editor Comments (if provided):

The effort toward improving the text led to a significant improvement of the manuscript. Unfortunately, however, quality of most figures is still very poor, the images extremely blurred and the numbers difficult to visualize cause these ito be n most cases useless to the reader. Unless this problem is not correctly addressed and solved, the manuscript can not be accepted for publication.
---

## [Author Response · Author response to Decision Letter 1]

1 Sep 2021

We would like to thank the Editor and Reviewers for their comments. We have updated all the figures again, to ensure maximum clarity, and also subjected each individual figure to the Preflight Analysis and Conversion Engine (PACE) digital diagnostic tool, https://pacev2.apexcovantage.com/ to ensure that every individual figure meets PLOS requirements. We can assure you that all figures meet the requirements. If there is anything wanting in any figure, please let us know the figure identification and we will rectify it immediately. 

As there are no other comments in the Decision Letter, we are re-submitting the manuscript for publication. Thank you.

---

## [Editor Report · Decision Letter 2]

6 Sep 2021

PONE-D-21-08604R2

Stage-differentiated ensemble modeling of DNA methylation landscapes uncovers salient biomarkers and prognostic signatures in colorectal cancer progression

PLOS ONE

Dear Dr. Palaniappan,

Thank you for submitting your manuscript to PLOS ONE. After careful consideration, we have decided that your manuscript does not meet our criteria for publication and must therefore be rejected.

Specifically:

I am sorry that we cannot be more positive on this occasion, but hope that you appreciate the reasons for this decision.

Yours sincerely,

Alessandro Weisz

Academic Editor

PLOS ONE

Additional Editor Comments (if provided):

The A.s failed to take any action toward improving the figure outlay, that in my opinion is not suitable for a scientific publication.
---

## [Author Response · Author response to Decision Letter 2]

16 Oct 2021

>>>Under "Specifically", the following was provided:

"I am sorry that we cannot be more positive on this occasion, but hope that you appreciate the reasons for this decision."

No concrete reason has been provided for the reject decision. There is an Additional Editor Comments (if provided):

"The A.s failed to take any action toward improving the figure outlay, that in my opinion is not suitable for a scientific publication."

It is possible that the links to the high-resolution figures from within the built PDF are not being viewed.

However that may be, we have carried out a restructuring of the manuscript and especially the figures, as follows:

(i) Figures 1, 2, & 3 combined into one figure 

(ii) Figures 4 & 8 converted into violinplots, and combined into one figure

(iii) Figures 5, 6, & 7 combined into one figure

(iv) Figures 12 & 13 combined into one figure

(v) Figure 14 replaced with Upset plot.

(vi) Figures 15 & 16 converted into violinplots and combined into one figure

The updated figures have again been checked with the PACE digital diagnostic tool (and run by third party readers). 

A detailed response is provided in the Response to Reviewers document and is reproduced here for convenience: 

>>> (1) Each figure has been subjected to the PACE digital diagnostic tool, and adjusted accordingly. All figures passed the PACE test. This is the standard used by PLOS ONE for all figures. How could the figures then be of poor quality? All figures are publication quality figures directly obtained from analysis software /algorithms. Hence the decision is a clear deviation from PLOS ONE’s editorial policy. 

>>> (2) We have complied with all the suggestions made to us with respect to our submission. As a proactive measure, we have reworked the entire set of figures and replaced it with a new more compact set of figures. This has been done in the following manner: 

(i) Figures 1, 2, & 3 combined into one figure 

(ii) Figures 4 & 8 converted into violinplots, and combined into one figure

(iii) Figures 5, 6, & 7 combined into one figure

(iv) Figures 12 & 13 combined into one figure

(v) Figure 14 replaced with Upset plot.

(vi) Figures 15 & 16 converted into violinplots and combined into one figure

This resulted in 11 figures from the original 19 figures. The reworked figures have again been checked with the PACE digital diagnostic tool (and run by third party readers). The manuscript has been accordingly updated. We request the reviewers to link to the high-resolution figures from the manuscript pdf.

>>> (3) To reflect all tracked changes since the original manuscript submission, the changes have been color-coded in the following manner: 

blue for revision-1, red for revision-2, and green for changes post appeal. 

Revision R2: Response to Reviewers:

Academic Editor: “The effort toward improving the text led to a significant improvement of the manuscript. Unfortunately, a however, quality of most figures is still very poor, the images extremely blurred and the numbers difficult to visualize cause these ito be n most cases useless to the reader. Unless this problem is not correctly addressed and solved, the manuscript can not be accepted for publication.”

>>>We would like to thank the Editor and Reviewers for their comments. We have updated all the figures again, to ensure maximum clarity, and also subjected each individual figure to the Preflight Analysis and Conversion Engine (PACE) digital diagnostic tool, https://pacev2.apexcovantage.com/ to ensure that every individual figure meets PLOS requirements. We can assure you that all figures meet the requirements. If there is anything wanting in any figure, please let us know the figure identification and we will rectify it immediately. 

Revision R1: Response to reviewers

>>>At the outset, we would like to thank the reviewers and the Editor for their valuable comments.

Academic Editor: The manuscript has been reviewed by two experts in the filed that both found it quite good and of interest, despite some problems, highlighted in particular by R.2, that must be addressed before it can be considered for publication.

>>>We have addressed the points raised by the reviewer#2 and have substantially revised the manuscript and expanded the scope of our investigations / discussion.

>>> Thank you, we have done the same. 

>>> Thank you, this has been done.

Reviewer #1: In this paper Muthamilselvan et al., developed a comprehensive computational framework for stage-differentiated modelling of DNA methylation landscapes in CRC, found significant changes and discovery a novel CIMP-like signature bearing potential clinical significance.

The data are supported by a strong statistical analysis.

The paper can be acceptted for pubblication.

Minor point

+Page 3 the word "Tomczak" must be deleted.

>>> Thank you, it was inadvertent and has been deleted. We would like to thank Reviewer #1 for their time and the kind comments.

Reviewer #2: In this study, the Authors propose a computational workflow for the analysis of the DNA methylation aberrations in colorectal cancer (CRC) with a stage-differentiated perspective. Data have been collected from The Cancer Genome Atlas (TCGA) portal; as a result, the Authors identify 7 stage-characteristic genes that could be indicative of a novel CpG island methylator phenotype (CIMP). The manuscript provides an interesting perspective and a detailed explanation of how DNA methylation data could be analyzed to detect epigenetic signatures between normal and tumoral samples or in different stages of the disease. Bioinformatic procedures are well described and depicting a useful workflow when handling big and complex data from repositories such as TCGA. Unfortunately, however, I cannot avoid pointing out that the work has serious shortcomings that do not allow to accept its publication in the present version and must be mandatorily corrected. Importantly, even if interesting, data are presented in a confused manner; in particular, the Authors should better define the aim of the study and the experimental design, carefully organize the Results, improve the Discussion and avoid exaggerated conclusions, concerning in particular inferences drawn from data that should be better supported by experimental validation. Specific comments are listed below.

>>> We would like to thank Reviewer #2 for the careful reading of our paper and the criticalcomments. We have addressed all the many valid points in the present revision. We have undertaken a major revision of the manuscript in line with the suggestions.

1. The Authors should carefully revise the text and correct some grammar mistakes.

>>> yes

2. Figures are in many cases blurred and not legible; their quality must be improved.

>>> all figures have been redone in high-resolution tiff format

3. The Authors should pay attention to some typing mistakes and font differences (see, for example, pages 14 and 12).

>>> yes

4. TCGA collects data from at least 10 different studies on CRC; at least the cohort from which data have been collected should be reported in Materials and Methods.

>>> this has been identified and recorded in the manuscript under Methods.

5. TCGA collects data from 236 patients profiled with Human Methylation Bead Chip HM27, and 393 with HM450, measuring 27,000 and 480,000 CpG sites, respectively; why only the HM27 data have been used?

>>> This point has also been addressed in the Methods section. Essentially 450k Chips show enrichment in gene body and intergenic regions. A distribution of the CpG sites with respect to the genomic / genic location clearly indicates this (data not shown). 27k data are enriched in CpG sites in promoter regions. Please refer the section under Methods.

6. The correlation analyses between methylation and gene expression data are providing interesting information but should be better described by focusing the attention not only on the methodological procedure used but also on the biological meaning of the observed results.

>>> This has now been rectified. Indeed, we now show plots for only the stage-salient genes, to make the biological connections and meaning clear. We note that all the results from our investigations are available in the Supplementary Files.

7. In the Conclusions the Authors write: “All the stage-salient genes were found to be hypermethylated, indicating a novel CIMP-like character possibly promoting epigenetic destabilisation, which in turn would drive the progression of colorectal cancer”. First, it is not clear where the hypermethylation associated with these stage-salient genes is located (promoter, TSS, CpG island or gene body); this should be better explained. Then, the role of the stage-salient genes identified by the Authors should be better characterized in the context of CRC to indicate a possible novel CIMP-like phenotype; I would suggest the Authors to enrich the Discussion by adding more details and experimental evidence of the involvement of these genes in CRC pathogenesis.

>>> We have now increased the literature weight for these statements and discussion. We have included a new Table 8 with the location of the DM probes. and a new Figure 19 to support these assertions. We further found a new publication citing stage-IV specificity for FAM123A while this manuscript was under review (medrxiv preprint of our work was available in October 2020). This has been included in the References. 

>>> Thank you.

---

## [Decision Letter · Decision Letter 3]

12 Jan 2022

PONE-D-21-08604R3Stage-differentiated ensemble modeling of DNA methylation landscapes uncovers salient biomarkers and prognostic signatures in colorectal cancer progressionPLOS ONE

Dear Dr. Palaniappan,

Thank you for submitting your manuscript to PLOS ONE. After careful consideration, we feel that it has merit but does not fully meet PLOS ONE’s publication criteria as it currently stands. Therefore, we invite you to submit a revised version of the manuscript that addresses the points raised below by the reviewers during the review process. In addition, the authors are requested to provide high quality figures.

We look forward to receiving your revised manuscript.

Kind regards,

Surinder K. Batra

Academic Editor

PLOS ONE

Journal Requirements:

Additional Editor Comments (if provided):

Reviewers' comments:

Reviewer's Responses to Questions

**Comments to the Author**

1. If the authors have adequately addressed your comments raised in a previous round of review and you feel that this manuscript is now acceptable for publication, you may indicate that here to bypass the “Comments to the Author” section, enter your conflict of interest statement in the “Confidential to Editor” section, and submit your "Accept" recommendation.

Reviewer #3: All comments have been addressed

Reviewer #4: (No Response)

2. Is the manuscript technically sound, and do the data support the conclusions?

Reviewer #3: Yes

Reviewer #4: Yes

3. Has the statistical analysis been performed appropriately and rigorously? 

Reviewer #3: Yes

Reviewer #4: Yes

4. Have the authors made all data underlying the findings in their manuscript fully available?

Reviewer #3: Yes

Reviewer #4: Yes

5. Is the manuscript presented in an intelligible fashion and written in standard English?

Reviewer #3: Yes

Reviewer #4: Yes

6. Review Comments to the Author

Reviewer #3: This is an interesting and potentially useful study where a comprehensive computational framework for modelling of the stage-associated DNA methylation in colorectal cancer (CRC) was developed. The authors have addressed previous critiques.

Reviewer #4: While the authors have addressed few of the comments from the previous revisions, a few minor points mainly about the cohort remain to be addressed in a greater detail-

1. As pointed out previously, there are various TCGA CRC the authors however do not clearly mention the details of the cohort.

2. It is briefly mentioned how the samples were from studies from Johns Hopkins or UCSC studies but it is unclear if these are two different studies from these centers (hence raising a question of batch effect) or if this was a combined effort. The authors should consider adding more information about the cohort.

7. PLOS authors have the option to publish the peer review history of their article (what does this mean?). If published, this will include your full peer review and any attached files.

Reviewer #3: No

Reviewer #4: No

---

## [Author Response · Author response to Decision Letter 3]

21 Jan 2022

Response to Reviewers:

>>> We would like to thank the Academic Editor and the Reviewers for helping improve our manuscript. All figures have individually passed the PACE digital diagnostic tool, https://pacev2.apexcovantage.com/ to ensure meeting PLOS requirements. References have been thoroughly checked once again.

Reviewer #3: This is an interesting and potentially useful study where a comprehensive computational framework for modelling of the stage-associated DNA methylation in colorectal cancer (CRC) was developed. The authors have addressed previous critiques.

>>> Thanks.

Reviewer #4: While the authors have addressed few of the comments from the previous revisions, a few minor points mainly about the cohort remain to be addressed in a greater detail-

1. As pointed out previously, there are various TCGA CRC the authors however do not clearly mention the details of the cohort.

2. It is briefly mentioned how the samples were from studies from Johns Hopkins or UCSC studies but it is unclear if these are two different studies from these centers (hence raising a question of batch effect) or if this was a combined effort. The authors should consider adding more information about the cohort.

>>> The comments are related, and we would like to thank the reviewer for raising this. The 27k methylation data were all shipped and processed by a single organization with code:05 JHU_USC center (Johns Hopkins – Univ. Southern California). This ensures homogeneity in data processing and submission to the TCGA. But the tissue source sites could be numerous (see below), so we carried out an analysis of batch-effects (https://bioinformatics.mdanderson.org/public-software/tcga-batch-effects/).

This yielded a Dispersion Separability Criterion of 0.299, which is < 0.5 and much lower than the recommended threshold of 1.0 for batch-correction. At the minimum, DSC values need to be >0.5 to consider the possibility of batch effects existing in the data. Thus, the samples within batches are as homogeneous to each other only as the batches themselves are to each other.

For instance, there were 29 tissue source sites for COAD alone:

1. Albert Einstein Medical Center

2. Mary Bird Perkins Cancer Center - Our Lady of the Lake

3. Duke University

4. University of Sao Paulo

5. Christiana Healthcare

6. Indivumed

7. International Genomics Consortium

8. Cureline

9. St. Joseph's Medical Center-(MD)

10. UNC

11. University of Pittsburgh

12. ILSBio

13. Harvard

14. MSKCC

15. Greater Poland Cancer Center

16. University Of Michigan

17. Asterand

18. Roswell Park

19. Candler

20. BLN - Baylor

21. University of Chicago

22. CHI-Penrose Colorado

23. Northwestern University

24. St. Joseph's Hospital AZ

25. Medical College of Georgia

26. Molecular Response

27. Institute of Human Virology Nigeria

28. University of Kansas

29. Wake Forest University

>>>We would like to thank the AE and Reviewers again for their comments.

---

## [Editor Report · Decision Letter 4]

2 Feb 2022

Stage-differentiated ensemble modeling of DNA methylation landscapes uncovers salient biomarkers and prognostic signatures in colorectal cancer progression

PONE-D-21-08604R4

Dear Dr. Palaniappan,

We’re pleased to inform you that your manuscript has been judged scientifically suitable for publication and will be formally accepted for publication once it meets all outstanding technical requirements.

Kind regards,

Surinder K. Batra

Academic Editor

PLOS ONE
---

## [Editor Report · Acceptance letter]

4 Feb 2022

PONE-D-21-08604R4 

Stage-differentiated ensemble modeling of DNA methylation landscapes uncovers salient biomarkers and prognostic signatures in colorectal cancer progression 

Dear Dr. Palaniappan:

I'm pleased to inform you that your manuscript has been deemed suitable for publication in PLOS ONE. Congratulations! Your manuscript is now with our production department. 

Kind regards, 

on behalf of

Prof. Surinder K. Batra 

Academic Editor

PLOS ONE